# Real-Time Augmented Reality Physics Simulator for Education

## Nak-Jun Sung [1], Jun Ma [1], Yoo-Joo Choi [2] and Min Hong [3],*

[1]   Department of Computer Science, Soonchunhyang University, Asan 31538, Korea; njsung@sch.ac.kr (N.-J.S.); majun@sch.ac.kr (J.M.)
[2]   Department of Newmedia, Seoul Media Institute of Technology, Seoul 07590, Korea; yjchoi@smit.ac.kr
[3]   Department of Computer Software Engineering, Soonchunhyang University, Asan 31538, Korea
*   Correspondence: mhong@sch.ac.kr

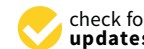

**Featured Application: The proposed technologies can be applied to the physics education system that describes the properties of materials, AR-based game engine to enhance realism, 3D product viewer engine for online shopping, etc.**

**Abstract:** Physics education applications using augmented reality technology, which has been developed extensively in recent years, have a lot of restrictions in terms of performance and accuracy. The purpose of our research is to develop a real-time simulation system for physics education that is based on parallel processing. In this paper, we present a video see-through AR (Augmented Reality) system that includes an environment recognizer using a depth image of Microsoft's Kinect V2 and a real-time soft body simulator based on parallel processing using GPU (Graphic Processing Unit). Soft body simulation can provide more realistic simulation results than rigid body simulation, so it can be more effective in systems for physics education. We have designed and implemented a system that provides the physical deformation and movement of 3D volumetric objects, and uses them in education. To verify the usefulness of the proposed system, we conducted a questionnaire survey of 10 students majoring in physics education. As a result of the questionnaire survey, 93% of respondents answered that they would like to use it for education. We plan to use the stand-alone AR device including one or more cameras to improve the system in the future.

**Keywords:** augmented reality; real-time physics simulator; Kinect V2; physics education

## 1. Introduction

Augmented reality is a technique of rendering a virtual object on a real image captured through a camera, and has the advantage of providing a sense of reality that is different from virtual reality. Recently, many augmented reality contents are being released due to the development of various augmented reality platforms (Vuforia, AUGMENT, et al.) and libraries (ARKit, ARCore, Maxst, et al.) [1–3]. Also, while Microsoft Hololens and Epson Moverio, which are dedicated hardware for augmented reality, have been unveiled, they are still unable to provide high-performance, due to low computing power [4,5]. Figure 1 shows Microsoft Hololens and Epson Moverio. Augmented reality is used in various fields, such as tourism, medical care, performance, education, and games. The AR technologies in particular have been actively applied to the fields of games and education.

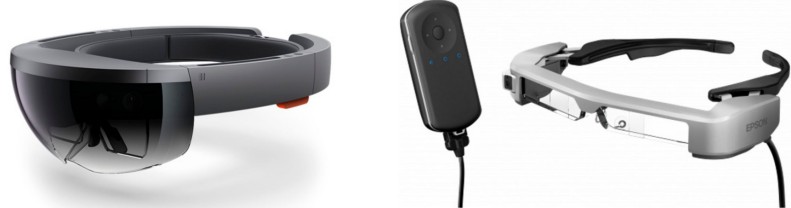

**Figure 1.** Recent Augmented Reality Devices (**Left**: Microsoft Hololens, **Right**: Epson Moverio).

Augmented reality in the education field has been actively researched since the 2000s, and many research results have been derived from both industrial and academic fields. Augmented reality content has the feature that users can directly interact with augmented 3D objects in the real world. Therefore, when providing AR-based educational content, learners can enjoy a more realistic and immersive experience than when using books or image-based education content. Especially in the field of education, physics can provide more realistic expressions through various physics-based simulation techniques, which enable students to gain high achievement.

Numerous studies have been carried out to express motion in the real world through physics-based simulation, such as free fall, parabolic motion, and comparison of changes according to object characteristics. However, it is difficult to apply AR-based physics simulation to education, because of the low computing power of the educational system. To solve this problem, we have developed a real-time simulator for the augmented reality environment based on lightweight physics simulation technology using parallel processing. The developed physical simulator simulates a deformable object for realistic presentation, and provides a form of the object in different shapes, according to various physical characteristics. We use Microsoft's Kinect v2 as hardware to provide the augmented reality, and the device has the advantage that it can later be converted to Microsoft Hololens V2.

The rest of the paper is organized as follows. Section 2 of this paper describes the research trends of augmented reality education content, and analyzes the physics education content in detail. Section 3 describes the design of the augmented reality simulator used in this paper, while Section 4 describes the implementation results of our content. Section 5 concludes the paper.

## 2. Related Works

### 2.1. Optical See-Through and Video See-Through

The techniques used in augmented reality are typically divided into optical see-through, and video see-through [6,7]. Figure 2 shows that the optical see-through system can simultaneously view the images rendered by the graphics system while viewing the real world through the optical combiner.

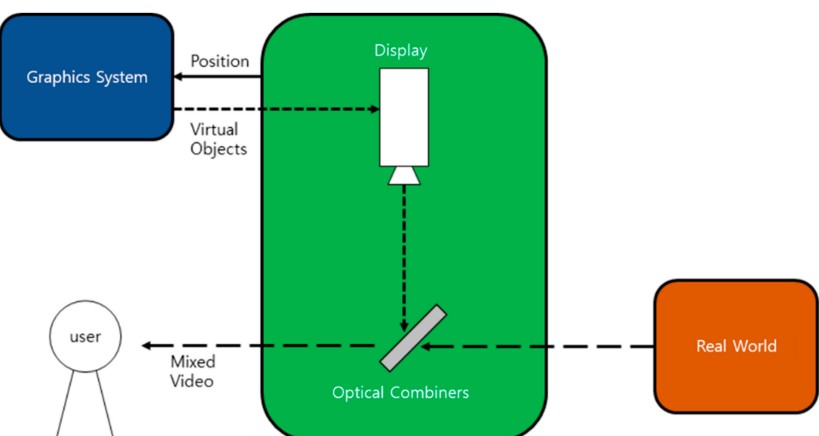

**Figure 2.** System Scenario of the Optical See-Through Method.

At this time, a marker-based AR for recognizing a marker through a camera or a non-marker recognition AR using a camera and a sensor outputs a virtual object to the optical synthesizing lens, after processing through the device's graphics system. The video see-through system is a method of acquiring real-world information by using one or more cameras as shown in Figure 3, synthesizing the information with the virtual information using the graphic system, and transmitting the information through the display. Most video see-through systems have the advantage of high-quality resolution and wide viewing angle, because they depend on the performance of the connected computer, not the stand-alone system. However, because of the way the camera displays information, there is a high probability of causing 3D nausea. On the other hand, most optical see-through systems are stand-alone devices with their built-in computer, which is specialized in portability and activity.

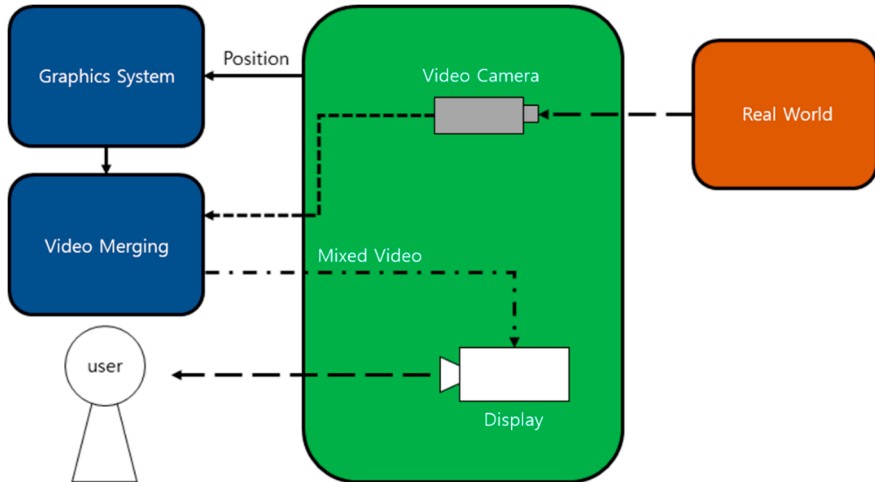

**Figure 3.** System Scenario of the Video See-Through Method.

The computing power of the optical see-through system is somewhat lower than that of the video see-through system based on the conventional desktop connection. However, it can be used while looking at the real world without the video camera. Therefore, when choosing the augmented reality implementation method, it is appropriate to select the video see-through method if the performance effectiveness is more important, or optical see-through method if the realistic visualization of the real world is more important.

### 2.2. AR with Physics Simulation

According to Chae et al., the research of physics simulation in the AR field is being carried out for the various purposes, such as expression of a virtual fluid in an augmented reality environment, a simulation of augmented reality ceramic production. Among them, several papers in the field of augmented reality physics simulation were reviewed [8]. Fujisawa et al. proposed a method of representing virtual fluid in the augmented reality environment. Smoothed Particle Hydrodynamics (SPH), a particle-based fluid simulation, was used to represent the realistic motion of the fluid, and marker-based augmented reality technology was applied. The simulation has the limitation that the marker is limited, and realistic simulation is difficult since the SPH method has low accuracy [9]. Leotta et al. have been working on creating 3D models automatically using non-aligned images. They created a 3D model by putting a virtual object on the screen representing the real world, and implemented the simulation of the interaction between the real object and the virtual object. The system provides many viewpoints to provide users with various simulation results, but the whole system performance depends on the amount of scene on the screen [10].

*2.3. Real-Time Physics Simulation*

Recently, a number of studies have been carried out to improve the performance of the real-time soft body simulation in the CPU environment and the GPU environment. Wang and Yang have developed a step length adjustment, initialization, and reversibility model conversion technique using GPU computing method to improve the performance of a deformed object simulator. In this case, the parallel processing algorithm was designed to reconsider memory efficiency, and the performance was improved by about 400 times using the GPU, compared to using the CPU [11]. Wang et al. presented an optimization method for high-resolution cloth simulation, and showed better performance for Newton's method and the projection dynamics method [12]. Our previous study evaluated the simulation performance of the CPU environment and the GPU environment through the experiment of freely dropping several 3D spheres, and showed that the performance of the GPU parallel processing environment is improved by about 84% [13,14].

For research on augmented reality using physics simulation, which is currently being researched and applied in a limited content, we are developing a novel augmented reality system of high performance using a video see-through method and a real-time physics simulator based on parallel processing using GPU.

## 3. System Information

In this section, we describe the configuration of the augmented reality environment and the real-time physics simulator for educational content. First, we describe the whole flow of the system proposed and implemented in this paper. We then describe the implementation method of each component constituting our system.

*3.1. System Flow*

The real-time augmented reality simulator for physics education is performed through the following system flow shown in Figure 4:

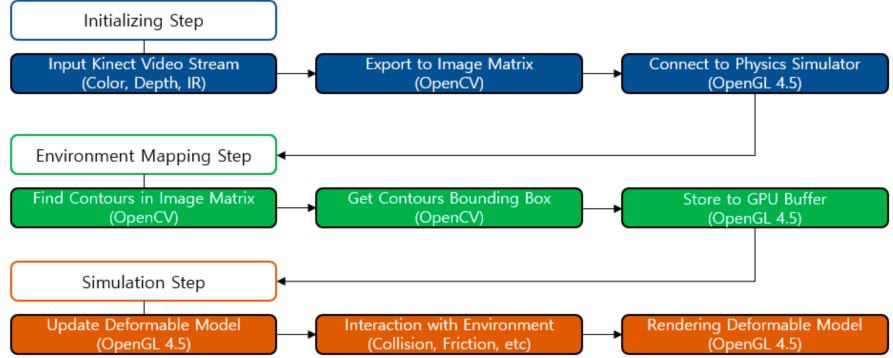

**Figure 4.** System Flow of the Real-Time Augmented Reality Physics Simulator.

Our system is divided into the initialization step → environment mapping step → simulation step sequence. In the initialization step, Kinect's video stream (color, depth, IR) is input for the augmented reality environment configuration. The received color and depth images have different resolutions, making it difficult to estimate the information in the environment mapping stage. To do this, the color and depth streams are calibrated in the initialization step. Then, each image is composed of an image matrix for each frame, and is connected to the physics simulator. The connected image matrix is used as a background for the augmented reality environment. In the environment mapping step, we use OpenCV, an image processing library, to extract meaningful information for interaction between the augmented object and the real world. First, in the step, all rectangles are found in the matched frame-specific image matrix. The rectangle is stored in the GPU buffer in the form of an Axis Aligned Bounding Box (AABB), which is one of the bounding boxes for smooth collision handling. After the

initialization step and environment mapping step are performed, object characteristics are set, and then the real-time simulation is performed, and interaction with the collision point of the detected real environment is performed.

### 3.2. Augmented Reality Environment Setup with Kinect V2

In this paper, we use a video see-through method to construct an augmented reality environment. We used Microsoft Kinect V2 camera for this purpose. Kinect V2 has the advantage of being able to simultaneously capture a color image, depth image, and IR image through multiple cameras, but calibration is required separately for analysis, using sensors of different resolutions [15]. Figure 5 shows the hardware configuration and information for Microsoft Kinect V2.

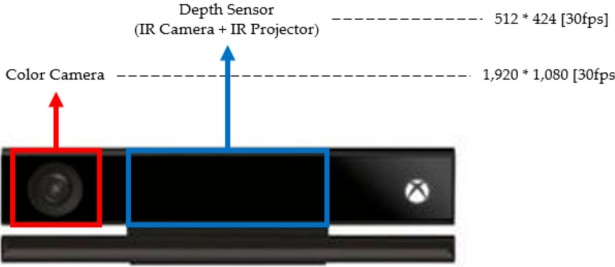

**Figure 5.** Microsoft Kinect V2's Hardware Structure Information (Resolution, Frame per Second (FPS)).

The image input through the color camera has a resolution of (1920 × 1080), but the image input through the depth sensor has a resolution of (512 × 424). We performed image calibration through the following algorithm for the analysis of two different images. The following Table 1 and Figure 6 show the image calibration algorithm and the result of our method:

**Table 1.** Color and Depth Image Matrix Calibration Algorithm for Image Analysis [16].

| Image Calibration Algorithm. |
|---|
| **Input: Color Image Matrix, Depth Image Matrix** |
| **Output: Calibrated Image Matrix** |
| //Preprocessing for Calibration |
| undistort color, depth image |
| create Calibrated Image Matrix for store calibration image [Depth.row * Depth.col] |
| |
| //Calibration Method (Loop Equation to Mapping Color & Depth Image) |
| for y = 0, y < Depth.row, y++ |
| for x = 0, x < Depth.col, x++ |
| float DepthValue = Depth(y,x) //get row, col depth value |
| vec3f DepthCoord = DepthToW(y, x, DepthValue) //create depth coordinate |
| vec2i ColorCoord = wToColor(DepthCoord) //depth coordinate to color coordinate |
| //Store color coordinate data to Calibrated Image Matrix (y,x) |
| Calibrated Image Matrix (y,x) = Color Image Matrix (ColorCoord[0], ColorCoord[1]) |

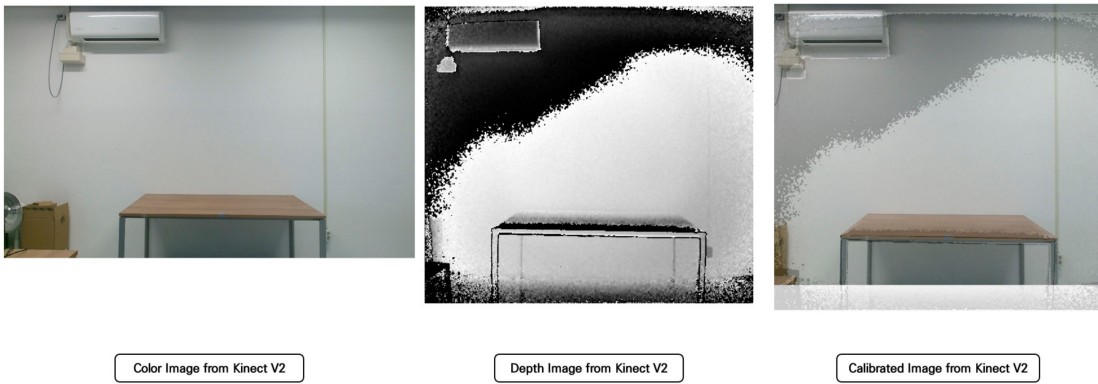

Color Image from Kinect V2     Depth Image from Kinect V2     Calibrated Image from Kinect V2

**Figure 6.** Result of Method (**Left**: Color Image, **Middle**: Depth Image, **Right**: Calibrated Image).

We derived the Calibrated Image Matrix from the matching algorithm, and analyzed the image of the matrix to extract meaningful information. Significant information, such as a desk or wall that can interact with the enhanced object, was detected in this study. The contours detection method provided by the image processing library OpenCV was used to detect (x, y) coordinates and (z) coordinates in the color image matrix, and to group them into one bounding box form. The information was then stored in the GPU buffer of the physics simulator to interact with the augmented object. Figure 7 shows the results of detecting squares using image processing from a video frame.

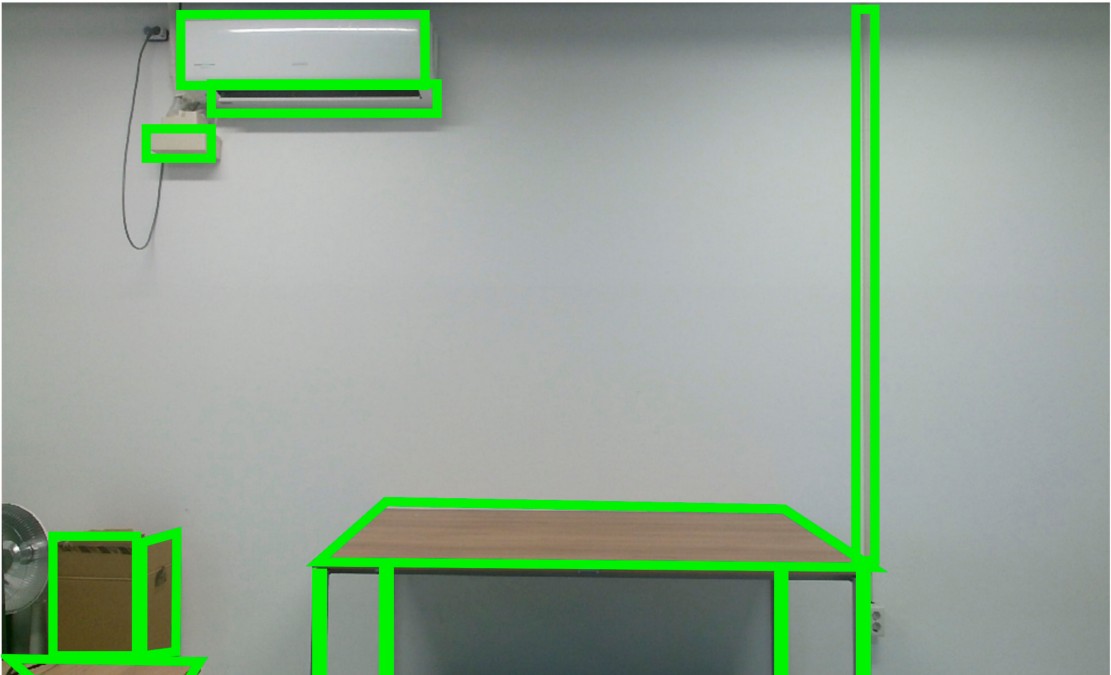

**Figure 7.** Find Squares in Video Frame (for Interacting with Physics Model).

### 3.3. Real-Time Physics Simulator

The physical simulator is divided into a rigid body simulator whose shape does not change when the force is applied, and a soft body simulator whose shape does change. The model on the left in Figure 8 shows after the collision of the rigid body, and the model on the right shows after the collision of the soft body.

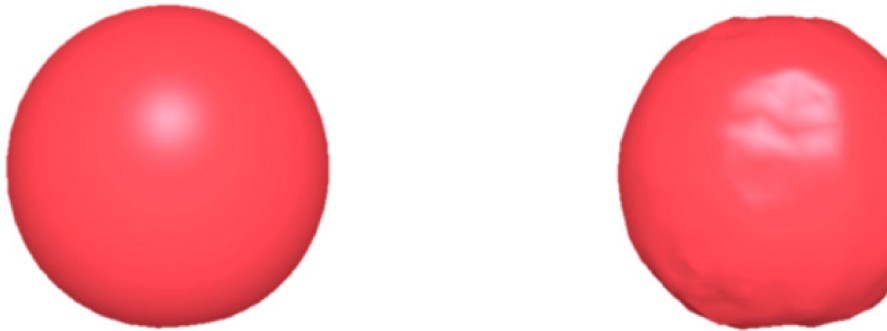

**Figure 8.** Behavior Comparison (**Left**: Rigid Body, **Right**: Soft Body).

The rigid body simulator has the advantage that the simulation speed is fast; however, the reality is low. The soft body simulator has the advantage of showing realistic motion; however, the simulation speed is slow. While most of the augmented reality content that performs simple augmentation uses rigid body simulators, it is preferable to use soft body simulators for content for physics education. Therefore, we developed a soft body simulator that can simulate real-time simulations, and applied it to augmented reality systems.

To perform soft body simulation, most researchers adopt deformation computing methods, such as the Finite Element Method (FEM), Mass Spring System (MSS), and Position Based Dynamics (PBD). Table 2 shows the advantages and disadvantages of each calculation method [17–19]:

**Table 2.** Comparison of Each Calculation's Advantages and Disadvantages (low, mid, high).

|  | FEM | MSS | PBD |
|---|---|---|---|
| **Simulation Speed** | low | **mid** | mid |
| **Simulation Accuracy** | high | **mid** | low |

FEM is chosen if precise simulation accuracy is required, while PBD is chosen if relatively fast speeds and inaccurate simulation accuracy are required. Without this extreme situation, most researchers choose MSS to perform soft body simulation. We implemented a soft body simulator by selecting MSS, which is a method of guaranteeing simulation accuracy and speed.

The MSS has the structure shown in Figure 9, in a way that the object is constituted by a point (node) having a mass and an imaginary line (spring) connecting it. When constructing a cloth model, all spring structures are used, but when constructing a volume model, only a structural spring is used.

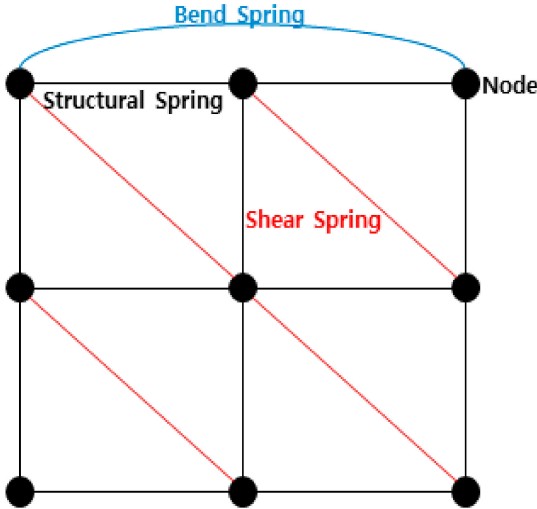

**Figure 9.** Structures of MSS (Node, Structural Spring, Shear Spring, Bend Spring) [20].

The MSS consists of a node and three springs (Structural, Shear, and Bend), and each component has the following data. The node has mass, position, velocity, and force basic data for each point. The spring has an initial length (*RestLength*), associated with the number of two connected nodes (*Index1* & *Index2*), a spring constant (*Ks*), and a damping constant (*Kd*) representing the characteristics of the spring as basic data. The MSS calculates the spring force through Hooke's Law based on the information of the node and the spring to express the strain. The force of the spring is used as an intermediary for tracing the position of the object through various numerical integration methods, such as the implicit Euler method, or the 4th order Runge–Kutta method. The deformation of the soft body is finally calculated through the prediction of the position of the object. Our soft body simulator satisfies at least 60 fps, and satisfies the constraints even when it performs in the augmented reality environment.

In this paper, we implemented a real-time MSS based on GPU parallel processing in the environment shown in Table 3.

**Table 3.** Development Environment (Real-Time Physics Simulator).

| NAME | DESCRIPTION |
|---|---|
| CPU | Intel i7 7700 3.60 GHz: 8 Core |
| GPU | Nvidia GeForce GTX 1080Ti |
| RAM | 32 GB |
| V-RAM | 11 GB |
| Library | OpenGL GLSL 4.3, OpenCV 2.4.9, Kinect V2 |
| IDE | Visual Studio 2013 Ultimate |

## 4. Implementation of the AR System

We implemented a real-time augmented reality physics simulator by combining the components described above. Through the implemented simulator, we simulated various objects with unique material properties. Our simulation models were generated by volume mesh model with sphere and torus shape using tetgen, a mesh generator. The following Table 4 shows the result of the wireframe rendering of volume mesh generated through tetgen [21]:

**Table 4.** Result of Generate Volume Mesh Model.

| Information | | Wireframe Rendering |
|---|---|---|
| **SPHERE** | | |
| NODE | 1.3 K | |
| FACE | 2.1 K | |
| SPRING | 25 K | |
| **TORUS** | | |
| NODE | 2 K | |
| FACE | 1.4 K | |
| SPRING | 62 K | |

In this research, the generated mesh models were simulated with a soft body simulator, and combined with an image system based on kinect v2 for augmented reality. First, we combined the real-time video stream with the soft body simulator for that. The following Figure 10 shows the combined result of kinect V2 and our soft body simulator:

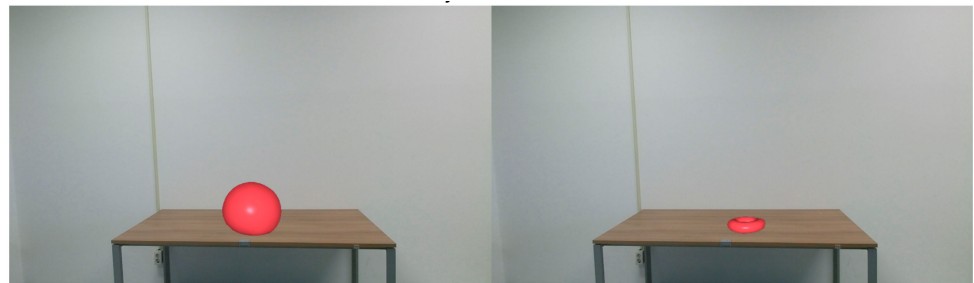

**Figure 10.** Combined Result of Kinect Video Stream and Our Soft Body Simulator.

Second, we created several objects with various material properties, and experimented with changes in object motion through simulation. The properties of various objects are expressed through the changes of the constant ($Ks$) and damping ($Kd$) of the object. $Ks$ is information that indicates the rigidity of an object, and even if it has the same outer shape, the force to maintain its shape becomes close to a rigid object when the value of $Ks$ is large. $Kd$ is the damping ratio that represents the attenuation of the force that causes the deformation of the object, which causes the object to stop, without moving forever. We also added frictional forces to the environment, to allow for more realistic movements to our system. The following Figure 11 shows the simulation results of different objects implemented through our system. We conducted an experiment, scene 1, that identifies differences in simulation results by applying different material properties, and an experiment, scene 2, that identifies differences in simulation results by applying different external forces.

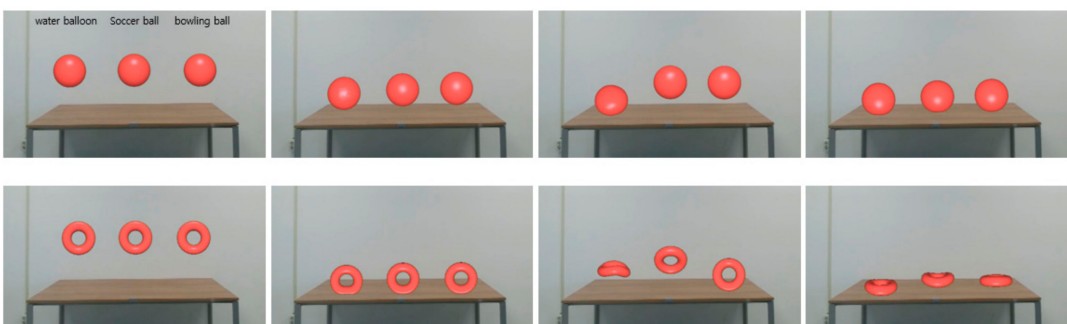

**Figure 11.** Result of Simulation Scene 1 (Freefall Simulation: Sphere and Torus Model).

We applied a water balloon, soccer ball, and bowling ball material to the sphere model by adjusting the material properties of each object. As a result of performing scene 1, we can see that the result of the simulation changes when different material properties are applied. The results of experiment 2 confirm that the deformation and the behavior of the object are different when different external forces are applied to the object having the same material properties. We applied external forces to the top of each of the three balls, and we compared the ball bouncing off the invisible wall. In the case of a stronger external force, the simulation was performed at high speed and the deformation was greater. Each snapshot in Figure 12 was taken every 1000 frames.

Finally, we conducted a survey to provide simulators to 10 students to teach physics, and to verify the educational effectiveness and responsiveness of the proposed system. Table 5 shows the items of our survey. We provided a total of five multiple-choice questions and two short-answer questions, and all students responded to the questionnaire [22].

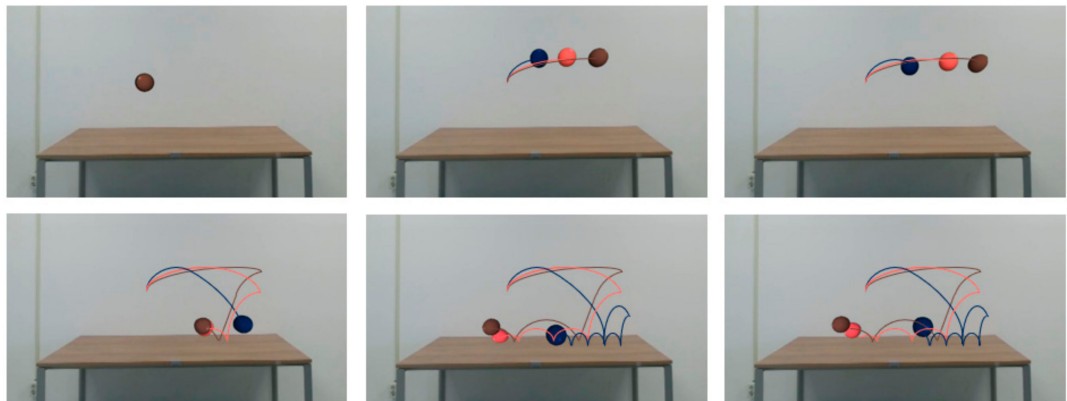

**Figure 12.** Result of Simulation Scene 2 (Apply difference External Force, Same Material Properties: Soccer Ball with movement trajectory).

**Table 5.** Our Survey Items (5 Multiple-Choice Questions [score 1: negative, 2: moderate, 3: positive], 2 Short-Answer Questions [doesn't have score]).

| Number | Question Detail |
|--------|-----------------|
| **Multiple-Choice Question** | |
| Q1 | I have encountered augmented reality. |
| Q2 | It is more interesting than a general book and video. |
| Q3 | There was a sense of reality. |
| Q4 | It is likely to help physics learning. |
| Q5 | I want to download and apply the simulator right now. |
| **Short-Answer Question** | |
| Q6 | What impact do you think AR technology can have on physics education? |
| Q7 | Are there any improvements to our system? |

The results of the survey's multiple-choice question are shown in the Figure 13.

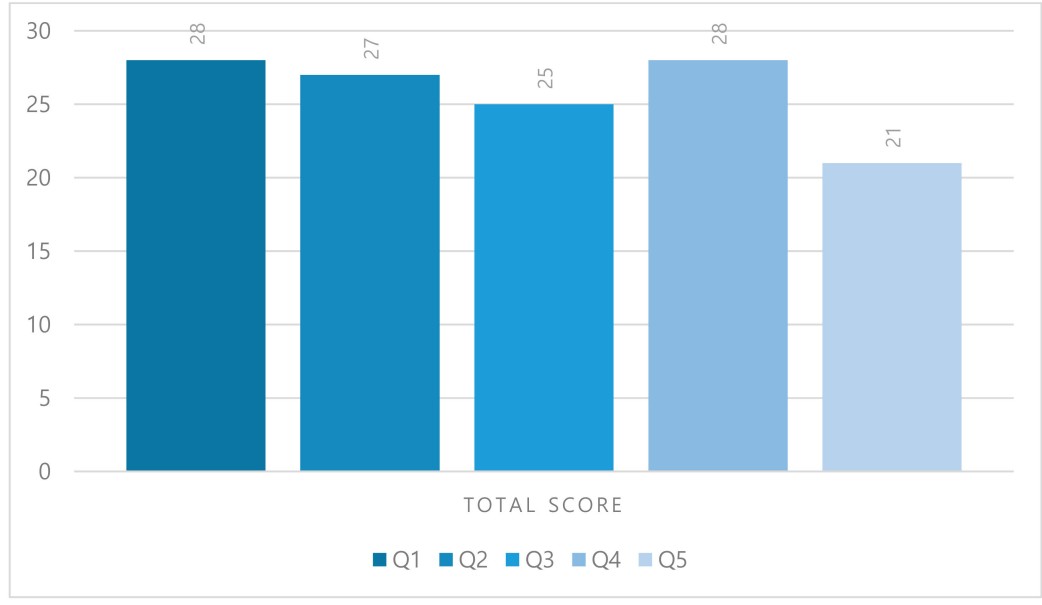

**Figure 13.** Result of the Survey.

Q1 and Q2 were answered most positively. However, there was a negative opinion that the Q5 would be difficult to use, because it required high specification and additional equipment. After conducting a short answer question, various opinions on the improvement of our system were

presented. There were positive feedbacks such as "I wish I could simulate various 3D objects" and "I wish I could make the simulation lighter and provide it on mobile.". Most of the answers to the Q6 were that mobile AR devices will be effective when commercialized.

## 5. Discussion and Conclusions

This research proposes a system that can provide more flexible educational content by providing real-time simulation technology for physics education, which is currently limited. We proposed physics education content by combining existing image processing technologies and our real-time soft body simulation technology. That system is implemented using Microsoft's Kinect v2, and has adopted the video see-through method. Augmented 3d model interaction with the real environment is possible by using basic image processing algorithms, such as a square detection algorithm.

After providing the simulator to 10 students who were learning to teach physics, the results of the questionnaire showed that 93% (28/30) of the respondents answered that the simulator will be helpful for physics education. The students surveyed commented that they expected to be more immersed and more focused through physics education using our simulator, than using a book and video. However, Q3 and Q5 had many negative answers compared to other questions. In particular, many commented that a limited environment should be created for augmented reality, and that it did not increase the realism much compared to books and videos. As a result of our survey, our simulator has the merit of providing more realistic representation than previous research, which will be helpful for education; however, it also has the disadvantage that it provides only a limited environment. For that, we want to provide more accessible content using high-end AR devices, such as Microsoft's Hololens V2, with the same image input mechanism. This research shows the result of the pilot research for this purpose. In future work, we plan to carry out research on the technology of providing physics simulation through the stand-alone AR device.

**Author Contributions:** M.H. provided conceptualization, project administration, and edited and reviewed the manuscript. N.-J.S. designed and implemented the software, and wrote the original draft. Y.-J.C. provided conceptualization and edited the manuscript. J.M. curated the data, and also wrote the original draft.

**Acknowledgments:** This work was supported by the Basic Science Research Program through the National Research Foundation of Korea (NRF-2017R1D1A1B03035718), funded by the Ministry of Education, and was supported by the Soonchunhyang University Research Fund.

**Conflicts of Interest:** The authors declare no conflict of interest.

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
