# Peer review of "Real-Time Augmented Reality Physics Simulator for Education"

_applsci, doi:10.3390/app9194019_

Round 1

Reviewer 1 Report

The authors proposed a video see-through AR system that includes an environment recognizer using a depth image of Microsoft's Kinect V2 and a real-time soft body simulator based on the parallel processing using GPU.
The abstract does not describe well about the proposed contents, particularly mentioned in the rigid/soft objects, because it is probably one of the main contributions in this paper.
The figures cannot make readers better understand about how the “see-through AR” works.
In the experimental results, the authors used questionnaire to evaluation the system. However, the questions are designed for general augmented reality and physical learning, not specifically designed for the proposed method. Therefore, the experiment can be redesigned to emphasize how the proposed augmented reality system impacts on physical learning.

Reviewer 2 Report

Interesting paper on Augmented reality applied to physics education. New perspectives are open to explain physical phenomena with AR. The experience is performed only for deformable object simulation, it would be interesting to simulated another physical problem to compare with this first trial. The reality sensation is not present enough, the physics teachers want maybe to demonstrate and measure, not only observe the deformed objects. The questions may be oriented to explore what the physics teachers need. The trajectory of the balls may be drawn to measure and compare with other trajectories.

The algorithm of calibration is not well explained. The English language can be improved. It would be nice to explain the content of the images from the experiment. Are the balls bouncing ? Maybe series of images can be more explicit.

Reviewer 3 Report

The main objective of this article is to develop a real-time augmented reality physic simulator for education using parallel processing techniques. Authors conducted a survey method to verify the usefulness of the proposed system. A high number of participants (students) responded that they would use the system. Furthermore, they present plans for improved system.

+Overall, it is a good and interesting research report.

-Needs a major editing for grammar and writing style.

Round 2

Reviewer 1 Report

The authors have revised the paper according to our comments given in the last version.
Some grammar mistakes are corrected in this revision.
The changes from the last version are listed as follows.
First, the authors have added the description of rigid/soft objects simulation to abstract, which is probably one of the main contributions in this paper.
Second, the authors adopt Figure 2 and Figure 3 for explanation of the “see-through AR” method.
Third, in the experimental questionnaire, the authors added two short-answer questions, which are “What impact do you think AR technology can have on physics education?” and “Are there any improvements to our system?”
Finally, the authors use the feedback obtained from the questionnaire to verify the effectiveness of their proposed method.
Although the experimental result is not very solid, it is acceptable.

Reviewer 2 Report

Dear authors, the article is significantly improved. I have two suggestions:

1) Figure 4 and in the text following this figure: Please write Initialization step or Initializing step instead of Initialize step. 

2) Figure 12: The movement trajectory is showing what happens in the image. You may add some sentences to explain: how the movement starts for the 3 balls (you put some force with what and in which direction ?); what happens next (they bounce against an invisible wall ?); and what is the time interval between each image.

Thanks.
